# Evaluating Feasibility of a Secondary Stroke Prevention Program

**DOI:** 10.3390/healthcare11192673

**Published:** 2023-10-02

**Authors:** Stephanie Hunter, Kimberley Vogel, Shane O’Leary, Jannette Maree Blennerhassett

**Affiliations:** 1Austin Health, Health Independence Program, Community Rehabilitation Service, Melbourne, VIC 3084, Australia; 2Austin Health, Spinal Community Integration Service, Melbourne, VIC 3101, Australia; 3Austin Health, Physiotherapy Department, Melbourne, VIC 3084, Australia

**Keywords:** secondary stroke prevention, exercise, exercise therapy, model of care, risk factors, health education, physical education and training, telerehabilitation, health risk behaviour, stroke

## Abstract

Healthy lifestyles including exercise and diet can reduce stroke risk, but stroke survivors often lack guidance to modify their lifestyles after hospital discharge. We evaluated the implementation of a new, secondary stroke prevention program involving supervised exercise, multidisciplinary education and coaching to address modifiable risk factors. The group-based program involved face-to-face and telehealth sessions. The primary outcomes were feasibility, examined via service information (referrals, uptake, participant demographics and costs), and participant acceptability (satisfaction and attendance). Secondary outcomes examined self-reported changes in lifestyle factors and pre–post scores on standardized clinical tests (e.g., waist circumference and 6-Minute Walk (6MWT)). We ran seven programs in 12 months, and 37 people participated. Attendance for education sessions was 79%, and 30/37 participants completed the full program. No adverse events occurred. Participant satisfaction was high for ‘relevance’ (100%), ‘felt safe to exercise’ (96%) and ‘intend to continue’ (96%). Most participants (88%) changed (on average) 2.5 lifestyle factors (diet, exercise, smoking and alcohol). Changes in clinical outcomes seemed promising, with some being statistically significant, e.g., 6MWT (MD 59 m, 95% CI 38 m to 80,159 m, *p* < 0.001) and waist circumference (MD −2.1 cm, 95%CI −3.9 cm to −1.4 cm, *p* < 0.001). The program was feasible to deliver, acceptable to participants and seemed beneficial for health. Access to similar programs may assist in secondary stroke prevention.

## 1. Introduction

Stroke is an episodic condition and a leading cause of disability and death worldwide [1,2,3,4]. In Australia, more than 445,087 people are living with the effects of stroke, often altering independence and quality of life, which impacts families, communities, healthcare and support services [1]. In Australia in 2020, 27,428 people had a stroke for the first time, with 24% of those people being under 55 years of age, and around 70,000 people were admitted to hospital for stroke [1]. The estimated economic impact of stroke in Australia in 2020 was AUD 6.2 billion for direct financial costs with an additional AUD 46 billion for mortality and loss of wellbeing [1]. These data also involve people who have recurrent stroke, which has an accumulative effect that increases the level of disability and demands on healthcare and further degrades quality of life [5].

The long-term risk and rate of recurrent stroke has been described as unacceptably high [6,7], especially given estimates that 80% of strokes can be prevented [2,8]. For example, a large longitudinal study in Canada found that people with stroke or transient ischaemic attack (TIA) who were clinically stable 90 days post-event had an 8-fold increase in hazard for recurrent stroke relative to matched controls at one year (HR 8.2, 95% CI 4.8 to 5.5) [6]. In Germany, a large population study found that the rate of recurrent stroke was 7.4% after 1 year and 19.4% after 5 years [9]. Therefore, there is an imperative to reduce rates of recurrent stroke [7]. Given that behavioural factors, such as exercise, diet and smoking, are estimated to account for 47% (41.3 to 54.4) of the burden of stroke [3], addressing those modifiable risk factors [7,10,11,12] has the potential to complement established medical and pharmacological management to reduce recurrent stroke [4,13,14]. For instance, a large population study found that people who were physically active had a 68% lower chance of stroke or death than people who were sedentary [15]. People with mild stroke or TIA have a 6-fold risk reduction for recurrent stroke if they undertake cardiovascular exercise, which is independent of receiving the recommended pharmacological management [16]. However, despite the health benefits of exercise, community-dwelling individuals with stroke are sedentary, spending the majority of their day sitting [17], a known risk factor for cardiovascular disease and stroke, including recurrent stroke [18].

Models of community care may support secondary stroke prevention [4,6], with guidelines recommending physical activity and cardiovascular exercise and referrals to support behaviour change to address modifiable risk factors [10,11,12,18,19]. Models of care, shown to be beneficial for health and fitness outcomes, include modified versions of cardiac rehabilitation programs [20]. The use of emerging technologies, such as telehealth and wearable devices, also has merit to support training, education, goal setting and monitoring to facilitate self-management programs [21]. Moreover, the use of telehealth has been feasible and may improve access to stroke care [22]. Factors that support people with stroke to be physically active and address lifestyle risk factors include guidance by health professionals who understand stroke, peer support and approaches that incorporate goal-directed and behavioural change [23,24]. These factors are consistent with the wishes of stroke survivors, who want more information on how to prevent stroke, including guidance on lifestyle and exercise [25,26], but these resources may not be provided routinely when people leave the hospital. When examining our local care in consecutive people with TIA, we observed that half of those people may get a brochure about the benefits of exercise, but few people were referred to health professionals to support behaviour changes for lifestyle factors (unpublished data), as recommended by current evidence-based guidelines to reduce the risk of a further stroke event [12,14,19]. These observations may be similar to other people with mild stroke who go directly home after a short hospital stay with medical and pharmacological management [27]. Given that current opinion and evidence cautions that the provision of simple advice or information about exercise may be ineffective [18], and that people with stroke or TIA may be unclear about safe and suitable exercise following a stroke event [18,23,25], we thought that providing access for community-dwelling people with stroke or TIA to modify lifestyle factors could support secondary stroke prevention. For the purpose of this study, secondary stroke prevention refers to interventions and strategies that aim to lower the risk of stroke recurrence for people who have had a stroke event, including TIA [10,12]. The program, outlined below, was designed to complement medical–surgical–pharmacological management.

Our primary aim was to evaluate the implementation of a secondary stroke prevention program provided within a Community Rehabilitation Service to see if it was feasible to deliver and acceptable for participants. The secondary aim was to evaluate if participation contributed to clinical outcomes that may help mitigate the risk of stroke. During the design and evaluation phase, we sought stakeholder, consumer and participant guidance to help ensure that the program met the needs of the participants. We hypothesised that we could design, deliver and implement a group-based program involving multidisciplinary education, supervised exercise and coaching to support people with mild stroke or TIA to modify lifestyle factors known to increase risk of further stroke. In addition, we hypothesized the program would be acceptable for participants.

## 2. Materials and Methods

### 2.1. Design and Setting

The study was an observational study that evaluated the implementation of a newly designed secondary stroke prevention program (during a 12-month period of intake, October 2021 to October 2022). The program was delivered within a community multidisciplinary rehabilitation setting, from the Health Independence Program at Austin Health, in Melbourne, Australia. Austin Health is a large metropolitan tertiary public health organization affiliated with universities for research and the teaching of medical, nursing and allied health. Austin Health admits around 500 people with a stroke event per year. Our stroke care includes an emergency department, an acute stroke unit, subacute inpatient and community rehabilitation and outpatient medical clinics. The project had ethical approval granted by the Austin Health Office for Research (reference number 20210118).

### 2.2. Participant Recruitment and Eligibility

People with mild stroke or TIA were invited to participate with the support of written participant information, discussion and opt-in consent. To be eligible, people needed to be community dwelling, over 18 years of age, within 4 months post stroke event, able to walk independently with or without an assistive device and living within the geographical area. Participants needed to be medically stable and have modifiable lifestyle risk factors for stroke. There were no additional exclusion criteria. People were referred to the program by allied health professionals, stroke liaison nurses or medical staff working in stroke care at Austin Health. This included acute wards, outpatient clinics (TIA and stroke prevention and rehabilitation physicians) and community rehabilitation. Eligibility and informed consent were confirmed via a triage process involving telephone contact with each person referred.

### 2.3. Program Format

The program was based on current guidelines [12,14,19] and comprised supervised exercise and multidisciplinary education to modify lifestyle risk factors for stroke. The program was delivered over a 12-week period and involved two phases: (1) supervised exercise and education and (2) coaching via telehealth to support self-management and behaviour change. Participants were assessed before starting to establish personal goals and preferences. Assessments were repeated midway (6 weeks) and at discharge (12 weeks) to provide feedback on progress, update the plan and support motivation. See Figure 1 and Table 1 below. The program was funded publicly at no cost to the participant. Participants were not provided with stipends.

Phase 1 (weeks 1 to 6, Figure 1): Participants had one 60 min session of supervised, group-based, onsite exercise per week for six weeks. (Group numbers were set by the COVID-19 mitigation strategies at the time. For the first intake, exercise was supervised via telehealth). The exercise program was designed and supervised by an Exercise Physiologist to ensure that exercise was safe, suitable and tailored to the individual’s ability, fitness and preferences. The key focus for exercise was moderate-intensity aerobic exercise, with coaching and guidance to support participants to learn how to feel safe and self-monitor their performance and intensity during exercise. The sessions also included discussion about behaviour change to problem solve strategies to overcome barriers for physical activity.

Phase 1 also included weekly education sessions (1 h duration) delivered in a group telehealth format (synchronous video using the Microsoft 365 Teams platform) by the multidisciplinary team from Austin Health’s Community Rehabilitation Service or Stroke Service (Table 1). All presenters were experienced in working with people with stroke. The series of education sessions was designed by relevant members of the multidisciplinary team with input from consumers, was adapted to support people with mild cognitive or communication difficulties and used existing trusted online resources [2]. Each session built upon and reinforced early components of the education. The educational model was interactive to enable peer discussion and support and promoted principles of behaviour change. All presenters received a detailed handover of the participants (medical, social and stroke-related impairments and goals) to promote relevance of each topic.

Phase 2 (week 7 to 12, Figure 1): Fortnightly telehealth coaching sessions (30 min to 1 h duration, synchronous video using the Microsoft 365 Teams platform) were undertaken and conducted by an Exercise Physiologist. These aimed to taper off the professional support, while encouraging the participants to self-manage their exercise/physical activity and lifestyle risk factors [18,21]. The coaching reinforced the person’s goals, checked in on their progress, and explored barriers and enablers to increase self-efficacy.

### 2.4. Outcome Measures

Our primary outcomes examined the feasibility and acceptability of delivering the secondary stroke prevention program. Feasibility considered the practicality of the new approach and was evaluated by collating a range of service delivery information, such as referrals (numbers and uptake), participant demographics and costs (staff time and wages). Participant acceptability was measured by participant uptake, attendance (number of sessions and proportion who completed the program) and satisfaction with the program. To determine satisfaction, we custom designed an online survey to enable participants to provide anonymous feedback about the program (using Microsoft 365 Forms). The survey focused on relevance, format (e.g., time and presentation), perceived safety and support and whether the program supported the ability to address lifestyle risk factors, as outlined by behaviour-change principles [28]. Survey responses included 5-point Likert scales and open text.

Our secondary outcomes involved clinical tests and patient-reported outcomes measured before commencing the program and at 12 weeks (discharge). The series of standardized clinical tests and validated questionnaires used are listed below. All clinical outcomes and questionnaires (with the exception of the Stroke Exercise Preference Inventory) were completed at initial assessment (before commencing program), midway (week 6) and discharge (week 12). Participants also provided self-reported information about changes in their lifestyle risk factors for stroke via the online, anonymous survey at the completion of the program (via Microsoft 365 Forms). The tests and questionnaires used are as follows:Blood pressure: measured using an automated machine (OMRON HEM-7203) with an inflatable cuff around the upper arm. For those with a stroke or TIA, blood pressures lower than 130/80 mmHg are recommended to reduce risk of recurrent stroke [29].Waist circumference (centimetres): measured at the level of the umbilicus with a tape measure being loose enough to fit one finger between the tape and participant. The Australian Heart Foundation recommends waist circumference less than 94 cm for males and less than 80 cm for females [30].Six-Minute Walk Test (6MWT): a widely used measure of functional walking endurance with high test–retest reliability, validity and established normative data for age and sex [31,32]. The test was conducted in a 30-metre corridor and included monitoring of cardiovascular parameters.Thirty-Second Sit to Stand Test: a practical test of functional leg strength and endurance with excellent test and retest reliability and validity. Scores reflect the number of times a person can complete sit to stand in 30 s from a 43 cm chair. Normative values in community-dwelling healthy adults aged 60–64 range from 12 to 17 for women and 14 to 19 for men [33].International Physical Activity Questionnaire (IPAQ) [34]: we used the short-form, 7-item IPAQ, which is a self-report of physical activity and sitting time over the past 7 days. The IPAQ has established test–retest reliability and validity, enabling estimates of total physical activity including vigorous intensity, moderate intensity and walking (in minutes per week) and time spent sitting (hours per week) [34].The Stroke Self-Efficacy Questionnaire (SSEQ) [35]: a self-reported questionnaire about level of confidence reported on a 0 to 10 scale (0 = not confident, 10 = very confident) when completing 13 activities of daily life following stroke. The SSEQ has good internal consistency and criterion validity. The overall score is the sum of all items.Fatigue Severity Scale (FSS) [36]: a valid and reliable scale to measure fatigue post stroke that involves rating agreement for 9 items on a 7-point Likert scale (1 = disagree, 7 = agree). Overall scores are averaged, reporting fatigue on a scale of 1 to 7, with higher scores reflecting higher levels of fatigue. The normal range is 2.3/7 or lower [37]. Scores higher than 4/7 reflect problematic fatigue in healthy adults [36].Stroke Exercise Preference Inventory (SEPI) [38]: a standardised questionnaire designed to explore preferences and barriers to exercise after stroke. Participants rate their level of agreement, where 0% represents ‘Don’t agree at all’ and 100% represents ‘Totally agree’. The SEPI was undertaken only at commencement to help establish participants’ preferences and understand perceived barriers to physical activity.

### 2.5. Data Analyses

Service information (referrals, referral uptake, participant demographics, staff costs to deliver the program and program details) were collated to examine if the program was feasible to deliver. Participant acceptability was summarized by collating information such as participant uptake, attendance and satisfaction. Clinical outcomes for group data were summarized (pre and post) and then analysed statistically using paired comparisons. We used a paired-sample *t*-test for interval data that were normally distributed and the Wilcoxon signed-rank test for non-parametric data or interval data that were not normally distributed. Statistical significance was set at 0.05. Paired comparisons were also reported as mean difference and 95% confidence intervals. Self-reported changes in lifestyle factors (provided in the online survey) were summarized descriptively outlining the proportion of participants who made lifestyle changes and the number of risk factors addressed. Data from the IPAQ (physical activity in minutes and time spent sitting per week in hours) were also collated. Statistical analyses were performed using R Statistical Software (version 4.2.2; R Core Team 2022). Online surveys and descriptive data were analysed using Microsoft 365 Forms and Excel. We did not undertake an intention-to-treat analysis, nor perform a priori sample size calculations.

## 3. Results

### 3.1. Feasibility

Over the evaluation period (intake October 2021 to October 2022), 90 referrals were received for the program. At triage, 37 people consented to participate, 10 were wait listed and 43 did not proceed with the program. The reasons for not participating were as follows: offered an alternative service such as 1:1 Exercise Physiology (28); medical (4); could not access telehealth (2); program was not indicated (3) and person’s choice (6). The 37 referrals that proceeded came from the acute ward (19), community rehabilitation (13), Better at Home (an inpatient bed substitution provided at home (3)), the Stroke Prevention Outpatient Clinic (1) and the external subacute rehabilitation setting (1).

Of the 37 participants, 73% were male, and the average age was 62 years. Most participants had an ischaemic stroke event (86%) and were a first presentation for a stroke event (86%). At triage, the average time since stroke event was 56 days. One-quarter of participants only received the secondary stroke prevention program, while the remainder also had Community Rehabilitation Services. For more demographic and medical information, see Table 2.

During the evaluation period, we delivered seven intakes of the program. Given that each intake had the capacity for eight participants, the overall uptake of the program during the evaluation period was 66%, while the later intakes ran at full capacity. No adverse events occurred. Seven participants did not complete the full program (defined by attending the final assessment session) for the following reasons: medically unwell unrelated to stroke (three); moved overseas (one); exacerbation of back pain (one); contracted COVID-19 (one) and returned to work (one).

We calculated that 122 staff hours were needed to deliver the 12-week program for eight participants to receive 24 occasions of service. These hours involved approximately 105 h of Exercise Physiology (for triage, assessment, education, coaching and general administration), with the remainder being for multidisciplinary education and administration. Based on 2023 award rates (including on-costs), the cost of the 12-week program was AUD 7760, and the cost per participant was AUD 970.

### 3.2. Acceptability

At triage, 83% of referrals wanted to participate in the 12-week program or an alternative service, and only 7% of people referred declined a service. Participants attended 79% of the telehealth education sessions and 83% of the supervised exercise sessions. The majority of the planned telehealth coaching sessions were completed. As mentioned previously, 81% of the participants completed the full program.

Twenty-four participants completed the online survey (65% response rate). The findings support that satisfaction was high (strongly agree and agree) for ‘relevance’ (100%), ‘would recommend to others’ (96%), ‘felt safe to exercise’ (96%) and ‘intend to continue’ (96%). See Table 3 for more details.

### 3.3. Clinical Outcomes and Stroke Risk Mitigation

The secondary outcomes involved clinical tests and questionnaires. Paired comparisons for the group show improvements after participation that were statistically significant for waist circumference, walking endurance, functional lower limb strength and levels of physical activity and fatigue. The changes observed for self-efficacy and blood pressure showed a diverse range and were not statistically significant (see Table 4).

Most participants (89%) reported that they had changed a lifestyle risk factor for stroke since attending the program (see Table 3). On average, each of those participants reported to have changed 2.5 lifestyle risk factors (diet, exercise, smoking and alcohol). Self-reports for physical activity (as reported by the IPAQ) indicated that participants increased physical activity time and reduced sedentary sitting time, and these changes were found to be statistically significant (see Table 4).

## 4. Discussion

The secondary stroke prevention program incorporating supervised exercise, multidisciplinary education and coaching was found to be feasible, safe and low cost to deliver. Participants found the program acceptable in terms of uptake, satisfaction and attendance. The clinical outcomes observed after participation also suggest possible benefits for addressing modifiable lifestyle factors associated with risk of recurrent stroke [3,4,10,14,18]. Our evaluation supports that this model of community care may be implemented within established multidisciplinary community teams with experience in stroke and links to acute stroke units to support people with mild stroke or transient ischaemic attack to address modifiable risk factors after leaving the hospital.

Elements of our program that may have assisted in making the model feasible to deliver and acceptable to participants require mention. The education, support and coaching were guided by evidence and principles of behaviour change [18,28]. The program employed skills and experience from a multidisciplinary team experienced in adapting programs for people with stroke [23,24]. Participants gave input to the program in terms of design, content and scheduling. During implementation, we also took on feedback from participants, staff and referrers to refine the service, making minor changes to streamline implementation (e.g., referral processes, patient information, etc.) and participant experience (e.g., provision of written handouts, emphasis for the education sessions and support to set up telehealth). This iterative process enabled us to document our model of care, permitting the possibility to share our procedures with other health services or to support research evaluation. Referral to the program was via trusted health professionals and followed up by a comprehensive triage discussion. Each person’s program was individualized for relevant risk factors, preferences for physical activity, physical capacity and goals. Feedback on progress was supported by the use of standardized tests, which included patient-reported outcomes. Participants also valued the interactive and supportive culture within the group-based education and supervised exercise sessions, as they provided opportunity for peer support and sharing of lived experiences. Having an initial phase of 6 weeks for telehealth and attendance also seemed palatable and practical, irrespective of the participant’s work, driving and social situation. The use of telehealth options also has scope to improve access, thus reducing impacts on carers and work schedules and overcoming barriers such as distance or transport [21,22]. It is, however, worth noting that some people required support from carers or staff to set up telehealth, leading us to recommend that future programs include options for onsite education. The inclusion of the second phase, with tapered coaching to enable the participant and clinician to communicate via telehealth, seemed helpful in reinforcing the importance of self-management for health promotion [14,18,21,26,28], while providing each patient with the opportunity and support to gain confidence to continue their chosen program.

In terms of feasibility, our program aimed to support people with mild stroke or TIA, as many of these people are not linked to health professionals for guidance to modify lifestyle risk factors for stroke, such as diet and exercise. Our referral data indicate that we captured 18% of the 500 people admitted annually to our health setting for stroke events. However, the low referral numbers for people with TIA or from the Stroke Prevention Outpatient Clinic highlights areas to target for referrals and service promotion. While our observation that few people (7%) declined a service for secondary stroke prevention may reflect a referral or selection bias, it seems that people want to improve their health outcomes and would benefit from the opportunity for support. Further, while the program ran at 66% capacity, it took time to promote a new service within our health setting and to establish clear referral pathways. At the end of the 12-month evaluation period, demand increased, and we now have a waiting list and continue to deliver the program as part of routine care.

Given that low levels of physical activity [2,3,12,17,18] and cardiovascular fitness [15,16,19] are risk factors for stroke, the observed gains for physical activity, walking endurance and lower limb function and the reduction in sitting time seemed promising. At discharge, the performance scores generally were within the normative range for the 6-Minute Walk Test [31,32] and the 30 s Sit to Stand Test [33]. On average, the group reported 4.6 h/day more physical activity and 1.6 h/day less sitting time. It was also promising that being more active did not seem to have a negative impact on fatigue. For instance, at commencement, the average level of fatigue reported approached that of being problematic [36] and improved towards normative values for healthy adults [37]. In terms of our other secondary outcomes, waist circumference scores showed improvements, but those changes were small and may not be clinically significant. No direct benefit for blood pressure was observed, but this was not surprising given that the participants were all medically stable, within the recommended range [29] and receiving pharmacological management.

The potential cost benefits of our program to reduce healthcare costs seem promising. Our 12-week program for eight people was estimated to cost AUD 7760 and may have contributed to reducing the risk of recurrent stroke given the observed increase in physical activity, walking endurance and changes in lifestyle. In comparison, the Australian weighted-average inpatient separation for stroke in 2020 costed AUD 10,209 [1], which suggests one hospital admission could fund 1.3 intakes of the secondary stroke prevention program. On approximate terms, four intakes of the program delivered to 32 people would have similar costs as three inpatient separations. The evidence supports that physical activity and exercise can reduce the risk of stroke (68% relative risk reduction [15] or reduced odds ratio of 0.4 (confidence intervals 0.2 to 0.6) [16]). Extrapolating from those data suggests that treating between two and five people (with exercise) may be needed to reduce one person from having a stroke. Given that recurrent stroke also has other economic and human costs [1,3,4,6,26], our program, which has been shown to be feasible to deliver and acceptable to participants, appears to offer good value for the money.

Several limitations of this study need consideration when interpreting and attempting to generalize from our findings. The study involved a small convenience sample of people from one healthcare setting with an established stroke unit and was from a high-income country. The participants had a range of medical conditions but were clinically stable and living independently in the community. Given the clinical nature of the program, the rate of attrition in attendance due to unrelated illness, relocation and return to work was not surprising, and seemed unrelated to the program. The study was also undertaken in the context of a program evaluation and was not designed to investigate the effectiveness of the intervention. The study was not controlled was under powered, and outcome measures were not undertaken by independent assessors. Moreover, the participants were aware of the program evaluation, and their positive views about the program were a source of potential bias. Further, while our approach did aim to change behaviour, we do not follow up participants to see if the observed changes were sustained. Given these limitations, the gains observed in the secondary outcomes should be interpreted with caution. Future controlled research is needed to determine if the program is effective, leads to sustained changes in lifestyle and reduces risk of recurrent stroke.

## 5. Conclusions

Our group-based program that combined supervised exercise, education and coaching to manage modifiable risk factors for stroke was feasible and low cost to deliver, acceptable to participants and may have supported beneficial clinical and health outcomes. This type of model of care has the scope to support people to address modifiable risk factors for stroke, and thus may assist in improving patient outcomes and optimizing the use of healthcare services. Further investigation of the program is needed to examine if it is effective and if changes in lifestyle are sustained.

## Figures and Tables

**Figure 1 healthcare-11-02673-f001:**
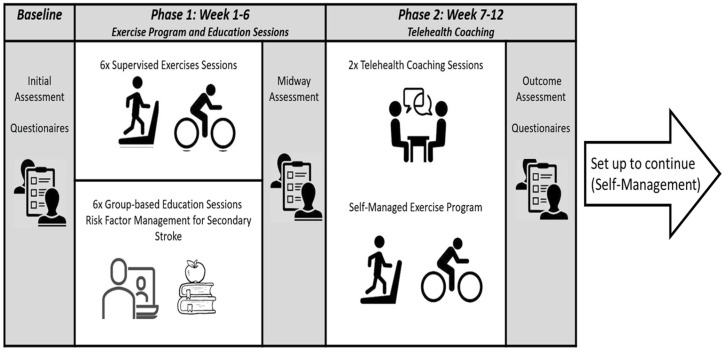
Infographic of secondary stroke prevention program showing assessment time points, phases and intent for continued self-management.

**Table 1 healthcare-11-02673-t001:** Education topics.

Week	Topic	Presenter ^1^
1	Welcome and overview of the program. Why should I exercise? How can I get started?	Exercise Physiologist
2	Overview of stroke and managing risk factors. Introduction to Stroke Foundation resources.	Stroke Liaison Nurse
3	Diet and Cholesterol.	Dietitian
4	Living Well after Stroke.	Occupational Therapy
5	How much exercise should I do?	Exercise Physiologist
6	Where to from here?	Exercise Physiologist

^1^ Senior staff from the Community Rehabilitation Service or Stroke Service.

**Table 2 healthcare-11-02673-t002:** Demographic and medical information about the 37 participants. Stroke event includes stroke or transient ischaemic attack.

Characteristics	Detail	Data
Age (years)	Average (SD)	62 (12) years
	Range	38 to 83 years
Gender (n)	Male	27
	Female	10
Type of stroke event (n)	Ischemic	32
	Haemorrhagic	2
	TIA	3
First vs. recurrent stroke event (n)	First stroke event	32
	Recurrent stroke event	5
Time since stroke event, at time of triage (days)	Average (SD)	55.9 (35.6) days
	Range, days	12 to 168 days ^1^
Relevant medical condition linked to stroke event (n)	Hypertension	9
	Carotid artery stenosis/occlusion	4
	Carotid artery dissection	2
	Intracranial radiotherapy	1
	Patent foramen ovale	4
	Thoracic aortic atherosclerosis	1
	Diabetic	5
	Atrial fibrillation	6
	Arteriovenous malformation	1
	Aneurysm	1
	No other relevant medical conditions	3
Mobility, (n)	Independent without gait aid	37
Required carer help to attend program (n)	No	36
	Yes	1
Community Rehabilitation Services provided (n)	SSPP only	9
	Community Rehabilitation Service and SSPP	28

Abbreviations: n = number, SD = standard deviation, TIA = transient ischaemic attack, SSPP = secondary stroke prevention program. ^1^ One person was referred 168 days post stroke, with the remainder meeting the 4-month selection criteria. This participant did not complete the program so their data are not included in pairwise comparisons.

**Table 3 healthcare-11-02673-t003:** Participant survey responses (n = 24 responses from 37 invited).

Survey Theme and Question	Percentage Overall Agreement ^1^	Breakdown of Responses (Percentage)				
		Strongly Agree	Agree	Neutral	Disagree	Strongly Disagree
General format of program						
The stroke prevention program was relevant to me.	100	79.2	20.8	0	0	0
I would recommend this overall program to other stroke survivors.	95.8	75	10.8	4.2	0	0
The time of the telehealth sessions suited me.	95.8	50	45.8	4.2	0	0
The format of the telehealth sessions suited me.	91.7	54.2	37.5	8.3	0	0
Relevance of each education session ^2^						
Exercise Physiology: Introduction to program. Benefits of exercise and how to get started.	100	62.5	37.5	0	0	0
Stroke Liaison Nurse: Overview of stroke and medical-pharmacological management.	95.8	54.2	41.7	0	4.2	0
Dietitian: Cholesterol and diet.	91.7	58.8	33.3	4.2	0	4.2
Occupational Therapy: Tips to resume life and activities.	91.7	62.7	29.2	4.2	4.2	0
Exercise Physiology: Tips to keep exercising.	100	79.2	20.8	0	0	0
Support to exercise						
I felt safe exercising at home.	95.8	66.7	29.2	4.2	0	0
There was enough follow up to help me check in on my progress.	95.8	75.0	20.8	4.2	0	0
Lifestyle and behavioural changes ^3^						
Since the program started, I have changed some lifestyle factors (e.g., diet, alcohol consumption, smoking, exercise).	87.5	62.5	25.0	12.5	0	0
I understand how much exercise I need to do to minimise my risk of secondary stroke. ^3^	100	66.7	33.3	0	0	0
I have the skills and resources to continue my exercise program long-term. ^2^	91.7	50	41.7	8.3	0	0
I am committed to continue my exercise to meet the secondary stroke guidelines and keep me healthy. ^3^	95.8	58.3	37.5	4.2	0	0

^1^ Data are summarized as overall agreement (responded strongly agree or agree). ^2^ Respondents were not asked to explain their rating of relevance. ^3^ Survey structured using COM-B model of behaviour change wheel: Capability (knowledge); Opportunity; and Motivation [28].

**Table 4 healthcare-11-02673-t004:** Observed mean and standard deviation for group data at commencement and discharge. Change scores recorded showing mean difference with 95% confidence intervals for the group.

Outcome Measure	Initial Assessment (n = 37)	Discharge Assessment (n = 30)	Paired-Comparisons Mean Difference, [95% CI] (n = 30) ^1^	*p*
Blood pressure (systolic) mmHg	125 (15)	124 (11.5)	−2.6, 95% CI [−8.4 to 3.2]	0.42
Blood pressure (diastolic) mmHg	79 (9.6)	79 (9.5)	−0.3, 95% CI [−3.7 to 3.0]	0.32
Waist circumference (cm)	100 (10.6)	98 (10.4)	−2.1, 95% CI [−2.8 to –1.4]	<0.001
6-Minute Walk Test (metres)	473 (83.6)	529 (88.6)	59, 95% CI [37.9 to 80.2]	<0.001
30 s Sit to Stand Test (repetitions)	13 (3.4)	15 (3.8)	2.4, 95% CI [0.9 to 4.0]	0.003
IPAQ total physical activity, (minutes per week)	350 (384)	582 (413)	276, 95% CI [−80 to 471]	0.008
IPAQ sitting time (hours per week)	6.2 (3.0)	4.5 (2.0)	−1.6, 95% CI [−3.1 to –0.7]	0.04
SSEQ (/130)	119 (9.6)	119 (10.1)	2.6, 95% CI [−2.5 to 7.7]	0.29
FSS (/7)	3.9 (1.5)	2.7 (1.2)	−0.9, 95% CI [−1.6 to 0]	0.08

Abbreviations: n = number, CI = confidence intervals, *p* = *p*-value, IPAQ = International Physical Activity Questionnaire, SSEQ = Stroke Self-Efficacy Questionnaire, FSS = Fatigue Severity Scale. ^1^ Data from the participant who was 168 days post stroke was not included in pairwise comparisons, as they did not complete the program for medical reasons unrelated to stroke.

## Data Availability

This was an observational study evaluating a program of care. The project did not require registration. The minimal datasets for this project are provided in the manuscript. Further data may be shared via correspondence with the authors.

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
