# Peer review of "Evaluating Feasibility of a Secondary Stroke Prevention Program"

_healthcare, 2023, doi:10.3390/healthcare11192673_

Round 1
Reviewer 1 Report
Dear Authors,
Stroke is the main reason for disabilities worldwide leading to motor dysfunction, cognitive problems, aphasia, and other pathologies, significantly reducing the quality of life. That is why the reviewed paper, evaluating the feasibility of a secondary stroke prevention program, is certainly a valuable research that brings novelty and assists to manage risk factors for stroke.
Please find below some recommendations that might help to improve the manuscript.
Introduction
The introduction section presents a significant background on stroke and recurrent stroke issues, models of care, etc. However, modern approaches to telerehabilitation after a stroke are not described. Moreover, the keywords contain “telerehabilitation”, however, the main text of the manuscript lacks this or similar terms. The paper might benefit if authors add a couple lines describing recent approaches to stroke rehabilitation using telemedicine, with relevant referencing.
Materials and Methods
Lines 120, 121 and 128, 129:
Readers will appreciate if authors mention the duration of education sessions (Phase 1) and telehealth coaching sessions (Phase 2).
If applicable, please specify a procedure for selecting a presenter for the data summarized in Table 1.
Also, the paper might benefit if authors mention any software they used to conduct the research and process the results.
Discussion
An important issue of the current research is that it aims to support stroke patients, as patients and healthcare professionals often lose touch with each other.
Maybe to introduce more discussion on stroke telerehabilitation as it improves patient-clinician communication, sustains patients’ compliance to treatment, including physical and cognitive exercises, lifestyle, provides access to healthcare and educational services for many post-stroke patients with immobility, living in remote areas, and during COVID-19 pandemic or similar events.
Please find below some papers that might help:
https://pubmed.ncbi.nlm.nih.gov/35527574/
https://pubmed.ncbi.nlm.nih.gov/36156443/
https://www.ncbi.nlm.nih.gov/pmc/articles/PMC9464491/
Relevant discussion on recent trends of stroke telerehabilitation as a part of a secondary stroke prevention might strengthen the paper making it more up-to-date attracting more readers, and stakeholders.
In general, the paper is the valuable research, which interesting to read, and can be considered for publication after revision.
Regards,
Reviewer
Author Response
Please find attached the updated review (to that submitted on 19/9). The updated review accommodated an additional review for our study.

Reviewer 2 Report
Thank you for the opportunity to review healthcare-2615588 manuscript “Evaluating feasibility of a secondary stroke prevention program”. The strengths of this manuscript are that it examines an issue of great public health relevance—stroke recovery and prevention in community settings. This project evaluates the feasibility, utility and acceptance of an Australian health system’s 12-month stroke-prevention education program for 37 (final sample 30) adult post-stroke/TIA participants (up to 4 months post-stroke). Seven 12-week sessions using telehealth and face-to-face formats structured into two phases were studied on primary outcome of feasibility, costs and secondary outcomes of multiple, standardized reliable and valid clinical outcome measures. The limitations of this manuscript are that the study design and statistics are not very clear due to missing methods components and areas of vague descriptions (study design, methods, primary outcome). Specific areas for follow-up and clarification are listed below. Thank you again for the opportunity to review this valuable topic.
General:
Since this is a research project with human subjects, the team should use the term ‘participants’ instead of consumers (i.e. L14, L18-19, L74, L76, L81 and throughout the entire manuscript, but especially in abstract and methods).
Some of the reference citations do not make sense for statistical justification of global stroke. For example, L40-41 should use global stats that are within the last 12-24 months and any key studies, not a 2017 commentary. Recommended examples:
World Stroke Organization (WSO): Global Stroke Fact Sheet 2022 (world-stroke-academy.org)
Global, regional, and national burden of stroke and its risk factors, 1990–2019: a systematic analysis for the Global Burden of Disease Study 2019 (thelancet.com)
· Note: Australia stroke authorities recommend most current WSO stats, and 2019 Global Burden of Disease study. Latest global stroke facts – Australian Stroke Alliance (austrokealliance.org.au)
Abstract
-If there are relevant primary outcome statistics that can be summarized, that would also be helpful.
-Stats summary says ‘not significant’ which implies hypothesis testing, but these results were not in the main text.
Introduction
-Summarize stroke as a leading cause of disability, and the most recent stats on that.
L51. Please confirm role of primary vs secondary stroke prevention programs/approaches, and provide a justification for emphasizing secondary in this study (since global recs emphasize primary prevention). Global, regional, and national burden of stroke and its risk factors, 1990–2019: a systematic analysis for the Global Burden of Disease Study 2019 (thelancet.com)
L60-64. This is confusing. Are these classified as primary or secondary program aspects for your org? For example, diet/exercise are usually part of the primary prevention toolbox, but if implemented post-TIA, post-stroke, does that make it secondary (d/t secondary tools are usually meds, surgical interventions, etc.)? Use of a formal stroke public health prevention framework could help here (i.e. US has CDC, AHRQ, American Stroke Association, etc).
Primary and Secondary Prevention of Ischemic Stroke and Cerebral Hemorrhage: JACC Focus Seminar | Journal of the American College of Cardiology
-Please insert information on the role of new science (i.e. genetics, AI) on stroke risk, since these are of increasing relevance to tailored prediction and prevention (Pharmacogenomics, Personalized Medicine, Precision Medicine, polygenic risk scores). Examples:
Stroke Risk Factors, Genetics, and Prevention | Circulation Research (ahajournals.org)
Stroke genetics informs drug discovery and risk prediction across ancestries | Nature
Precision medicine in stroke: towards personalized outcome predictions using artificial intelligence - PMC (nih.gov)
L81. If this is a research project, please insert a clear research question. Or if EBP/QI, a PICOT question, etc.
Methods
L84. Since there is IRB/ethics committee approval and survey instrument measurements, the manuscript ‘reads’ like an observational research study (also stated on p11, Data availability statement). The terms ‘12-month evaluation’, ‘consumers’, ‘clinical guidelines’ and ‘multiple program cycles’ (which are more on the EBP-QI ends of things) are confusing. For example, in the US, we use standardized frameworks for stroke program evaluations, such as the CDC, AHRQ. AHRQ often links them to specific indications, such as a-fib. Examples:
1) CDC: Introduction to Program Evaluation for Public Health Programs @ Introduction to (cdc.gov) and Program Evaluation Home - CDC and
2) AHRQ: Stroke Prevention in Atrial Fibrillation | Effective Health Care (EHC) Program (ahrq.gov).
The study design should be stated clearly for validity, transparency, replication, etc. For example, if I use the EQUATOR international reporting guidelines, what would this project be categorized as? QI, EBP (PICOTs) and CPGs? observational study, economic evaluation?
L87. Is this a tertiary or academic medical center affiliated with a school of medicine?
L92. Any exclusion criteria or key inclusion criteria? I.e. comorbidities (a-fib), medications, insurance benefits, language, etc. Any study participation ‘benefits’ such as meal compensations, session stipends? Sampling--convenience sampling from referrals?
L104. Should clearly state each guideline, version used for transparency. The 12-month program timeline should be in methods—any COVID-19 impacts? L120-123. Is there an Australian public health framework that was used to develop the stroke info education module (table 1), similar to the US’ CDC resources?
L139. Primary outcome measures: should clearly state the dependent variables being measured (clearly operationalized definitions and values, metrics).
L143-144. Error in table ordering, this should be Table 2.
L148. This secondary outcomes section is very well done.
L190. Several components are missing or underdeveloped: identify stats program, version# used, accurately state the names of the descriptive and inferential statistical tests (i.e. parametric, non-parametric), any power analyses, how missing data were handled, and/or if any insurance or public administrative databases were used for benchmark comparisons, etc.
Results
L245. I’m not following Table 4. ‘Pre, post-scores, comparisons’ are usually the language of paired t-tests. Shouldn’t this have a mean, SD, SEM then the 95%CI of the difference for the theoretical probability distribution used (i.e. t-distribution)? Can you please confirm.
-Confirm table formatting (left justified vs centered).
Discussion
L324. What in your opinion were the confounders that should be accounted for in subsequent research? Were there any key lessons learned from the attrition cases? Recommend addressing the typical language terms of generalizability, reliability, validity, bias, etc.
L329. Descriptive statistics? Unclear d/t pre/post language.
-Any similar stroke prevention programs in Australia for comparison? What makes this project innovative since heart disease and stroke prevention programs are typically bread/butter in many HC systems (traditional everyday activities)?
-Placing into context of primary vs secondary prevention programs here would be super helpful.
-Future research? I.e. grant funding, etc.
References
Formatting, typos. Recommend adding stronger science citations for most recent stroke stats from WSO, Lancet and science/tech trends. Try to strengthen references with the most recent available, i.e. last 2-3 ideal, 5-10 years acceptable, and/or consistency with the desired population of stroke patients (not another unrelated illness, i.e. parkinson’s, lupus).
Very good English language. However, need to increase the precision in key terminology for methods and other key areas (i.e. abstract stats, etc).
Author Response

(The authors gave the same response as above.)

Reviewer 3 Report
As articulated by the researcher, the prevention of secondary stroke is paramount. Consequently, a program dedicated to preventing secondary strokes emerges as an essential study.
*Introduction
This study ostensibly focuses on the program for the prevention of secondary strokes. However, the Introduction seems to lack references to other studies addressing secondary strokes. Highlighting statistical data, such as the percentage of secondary strokes in the overall stroke occurrences, would accentuate the necessity of this research.
*Materials and Methods
The program is commendably structured, comprising 6 weeks of on-site sessions followed by another 6 weeks of remote sessions.
*2.4 Outcome Measures
A clearer exposition regarding the instruments employed for measurements in the "Outcome measures" section seems warranted.
*2.2 Participant Recruitment and Eligibility
The eligibility criteria specify "within 4 months post stroke-event." However, in Table 2, under "Time since stroke-event, at time of triage, (days)," the range provided is "12 to 391 days." This suggests the inclusion of data inconsistent with the participant criteria — a serious discrepancy that cannot be overlooked.
*Table 3
Within the "Relevance of each education session" section, for the items "Disagree" and "Strongly disagree," it would benefit readers if specific reasons were elucidated for the negative responses associated with certain educational topics.
*Table 4
Regarding "Initial Assessment" and "Discharge Assessment," it's imperative to indicate whether the changes observed are statistically significant. As it stands, only score changes are presented, making it challenging to discern the efficacy of the secondary stroke prevention program executed in this study.
*Discussion
The mention of the cost-effectiveness of the program is indeed exemplary.
Author Response

(The authors gave the same response as above.)

Round 2
Reviewer 1 Report
Dear Authors,
Although the paper has been improved, there are issues that require revision, specifically, the reference list.
There are some typos with reference [22] (year of issue and publisher name):
Typo: 2023.
Correction: 2022.
Typo: Neuro Rehabil
Correction: NeuroRehabilitation
https://pubmed.ncbi.nlm.nih.gov/35527574/
(The full and abbreviated names of the journal "NeuroRehabilitation" are the same.)
A part of the title is missed in reference [9]:
Typo: “The frequency and timing of recurrent stroke,”
Correction: "The Frequency and Timing of Recurrent Stroke: An Analysis of Routine Health Insurance Data".
https://pubmed.ncbi.nlm.nih.gov/31711561/
Regards,
Reviewer
Author Response
Please find report attached.

Reviewer 3 Report
The completion quality of the manuscript, after statistical analysis and revisions, has notably improved. However, it appears that there are still some amendments to be addressed.
In the eligibility criteria, it is specified that participants should be "within 4 months post-stroke." However, the range provided in Table 2 under "Time elapsed after stroke, classification point (days)" is "12~168 days." I acknowledge the clarification provided on the initial misrepresentation and appreciate the additional information about the participants written below the table. Nevertheless, there is still data included that doesn't align with the criterion of "within 4 months post-stroke." Hence, it seems appropriate that data not fitting this criterion should be excluded and adjusted in the study's findings.
Further elucidation in the Discussion section is warranted regarding areas that showed statistically significant changes as a result of the interventions executed in this study, such as the increase in participants' physical activity. Specifically, a comparison between previous studies and the current study's results, like the unchanged aspect of blood pressure and the increased physical activity, would enhance the completeness and depth of the manuscript.
Author Response
Please find report attached
